# Energy Autonomous Wireless Sensor Nodes for Freight Train Braking Systems Monitoring

**DOI:** 10.3390/s22051876

**Published:** 2022-02-27

**Authors:** Federico Zanelli, Marco Mauri, Francesco Castelli-Dezza, Edoardo Sabbioni, Davide Tarsitano, Nicola Debattisti

**Affiliations:** Department of Mechanical Engineering, Politecnico di Milano, 20156 Milan, Italy; marco.mauri@polimi.it (M.M.); francesco.castellidezza@polimi.it (F.C.-D.); edoardo.sabbioni@polimi.it (E.S.); davide.tarsitano@polimi.it (D.T.); nicola.debattisti@polimi.it (N.D.)

**Keywords:** wireless sensor node, MEMS, energy harvesting, pressure measurements, brake system, freight transportation, monitoring, predictive maintenance

## Abstract

Nowadays, railway freight transportation is becoming more and more crucial since it represents the best alternative to road transport in terms of sustainability, pollution, and impact on the environment and on public health. Upgrading the potentiality of this kind of transportation, it would be possible to avoid delays in goods deliveries due to road accidents, traffic jams, and other situation occurring on roads. A key factor in this framework is therefore represented by monitoring and maintenance of the train components. Implementing a real time monitoring of the main components and a predictive maintenance approach, it would be possible to avoid unexpected breakdowns and consequently unavailability of wagons for unscheduled repair activities. As highlighted in recent statistical analysis, one of the elements more critical in case of failure is represented by the brake system. In this view, a real time monitoring of pressure values in some specific points of the system would provide significant information on its health status. In addition, since the braking actions are related to the load present on the convoy, thanks to this kind of monitoring, it would be possible to appreciate the different behavior of the system in case of loaded and unloaded trains. This paper presented an innovative wireless monitoring system to perform brake system diagnostics. A low-power system architecture, in terms of energy harvesting and wireless communication, was developed due to the difficulty in applying a wired monitoring system to a freight convoy. The developed system allows acquiring brake pressure data in critical points in order to verify the correct behavior of the brake system. Experimental results collected during a five-month field test were provided to validate the approach.

## 1. Introduction

As highlighted in recent EU-funded research projects, the performances of railway freight transportation have increased in the last few years and the projection is that the trend will continue in the future because it represents the best alternative from the sustainability point of view with respect to road freight transportation, which instead has a huge impact on the environment [1,2,3,4].

However, rail freight transport is still affected by issues that limit its usage and efficiency. An example of this is represented by derailments that, although decreasing in number in the last few years, still occur on tracks around the world, leading to a decline in confidence in this transportation mode [5]. 

The growing idea in the field is therefore to improve the reliability and performances of these vehicles by implementing smart and innovative technologies useful to carry out continuous monitoring of the main crucial components and to apply predictive maintenance logistics [6]. In this way, it could be possible to avoid expensive maintenance interventions and higher costs due to the unavailability of the wagon for operations. However, limited data have been collected so far on these vehicles for predictive maintenance purposes because of lack of appropriate instrumentation. Monitoring systems suitable to the purpose must in fact be characterized by low costs and easy installation on the wagon, by impacting the re-approval procedure as little as possible. Moreover, due to the lack of any power sources on board freight wagons, the use of energy harvesting techniques is necessary to supply the installed instrumentation [7]. 

A recurrent failure with a high impact on reliability and service is related to the braking system in which, if a malfunction occurs, it results in complexity in finding the cause of the fault within the system. In addition to vehicle failures, a possible common consequence of the malfunctioning of the braking system is the onset of wear on the rolling surface (facets). The presence of facets implies a damaging action of the traveling wagon on the infrastructure itself [8,9]. 

For these reasons, the braking system plays a fundamental role in traffic safety, and it has a great influence on the occurrence of anomalies on other components of the mechanical system (such as running gear, traction systems, etc.). This, therefore, translates into greater costs in case of breakdown and in the difficulty of identifying anomalies during routine maintenance operations on this type of vehicle. The problem has been recently widely studied more from the numerical point of view through the development of models able to reproduce the air brake system behavior [10,11] than from the experimental side, where few examples of experimental tests on the field can be found [12]. Clearly, the possibility of monitoring the braking system would represent a smart solution since it is present almost on all the wagons and would allow the recording of the braking actions according to the load and the line traveled. 

The air brake system of a freight train has a very similar structure among the various models present on the market. Every freight wagon is equipped with a brake main pipe, which connects the locomotive to the wagons. Downstream of the brake main pipe, it is connected the control valve which manages the auxiliary tank, the control tank, and the brake cylinder. When the driver activates the brake system, a pressure variation occurs in the main brake pipe because it is put in communication with the atmosphere or with the main tank. 

Another fundamental component of the brake system is represented by the weighing valve mounted on every bogie. The weighing valve allows regulation of the pressure of the brake cylinder by supplying a pressure value proportional to the load carried on the wagon. The pressure generated in the brake cylinder is almost proportional to the pressure drop in the main brake pipe and the performance characteristic depending on the transfer curve of the control valve. The control valve’s main task is to compare the pressure value in the main brake pipe with the control tank one. If a pressure drop in the main brake pipe is observed, the control valve put in communication the auxiliary tank with the brake cylinder and the braking phase is started. Moreover, the control valve has the role to guarantee the braking and release time according the UIC540 standard, which are different in case the braking mode is set to P (passenger train) or G (freight train) [13]. It is therefore evident that the monitoring of pressure trends in some crucial point of the braking system allows the identification of eventual malfunctioning of its main components and a smart solution for the monitoring of the braking system of freight wagons, as was proposed by [13]. 

In [13], the temperature of brake blocks and the pressure variations was measured in some key points of the braking plant, and the significance of the collected data and the efficiency of the developed monitoring systems was demonstrated. However, the main issues of this monitoring apparatus were represented by the presence of wired sensors and by the limited autonomy due to the lack of any energy harvesters. The previous problems underline that to improve system reliability it is necessary to install a smart monitoring system on wagons to perform predictive maintenance of the braking system. This kind of apparatus should take advantage of wireless communication due to the unfeasibility of wirings on this kind of vehicles and of energy harvesting since no power supply is nowadays present on freight wagons. Moreover, measurements should be performed in peculiar points of the system useful to identify possible failures of the main components, without undermining the safety of the braking plant.

Regarding the wireless communication, it has successfully employed on several kinds of monitoring activities on different mechanical and civil structures [14,15,16,17,18,19,20,21], while its use on rail vehicles is not common due to the presence of a huge quantity of iron that can affect the quality of transmission. The durability of this kind of sensor is also a concern because this monitoring activity surely represents a harsh scenario in terms of environmental actions. An example can be represented by the ballast lifting from the track bed that can impact instrumentation installed on the bogie [22]. 

The need for a low-energy consumption of the system is a priority in this context, and it can be reached both by adopting ultra-low-power electronics component and optimizing the acquisition cycles. The introduction of a low-consumption status when the acquisition is not running is of paramount importance to save unneeded power.

In the end, many kinds of energy harvesters have been tested and implemented in recent years to be used in monitoring systems with the aim to recharge batteries when an energy excess is produced [23,24,25,26,27]. Among them, solar energy is seen as the most reliable and widely available in outdoor scenarios [28,29]. It can be found in literature that the implementation of photovoltaic panels used to harvest solar energy on wireless sensor nodes resulted in successful application of vibration monitoring of a freight wagon [30].

For these reasons, an innovative system composed by wireless sensor nodes endowed of pressure transducers was developed to be used for the monitoring of pressure variation inside the braking system of a freight wagon. Sensor nodes were designed to be energetically autonomous thanks to the choice of low-power electronic components and to the development of an optimized state machine able to minimize energy consumptions. The developed system was then installed on a freight wagon for a 5-month field test. Sensor nodes were connected to crucial points of the braking plant in order to perform pressure measurements useful for continuous monitoring of the system.

The paper is organized as follows. Section 2 describes the design of the wireless sensor node, both from the hardware and software point of view. A laboratory calibration of the developed device is shown in Section 3, while Section 4 is dedicated to outline the field test and the arrangement of the experimental set-up. In the end, in Section 5, we propose an analysis of the experimental data collected with a focus on the consistency of the acquired data, on the energy harvester performance, and on the overall efficiency of the monitoring system. Finally, it is shown how the collected data can be used to pursue a predictive maintenance approach able to identify possible faults in the brake system. 

## 2. Sensor Description

The developed pressure wireless sensor node was designed to acquire brake system pressures over time and to perform some preliminary calculations in order to minimize the transfer of data reducing the power consumption. Indeed, the power supply represents a critical issue in this application due to the lack of train power line on freight convoys and the complexity of installing cables on existing wagon. For this last reason, wireless communication is also an essential element to overcome these limits, assuring a proper data transfer while keeping a low power consumption.

In this application, the sensor nodes were developed starting from the design of vibration sensor nodes described in [31]. The power source of the sensor nodes is essentially provided by a Lithium Polymer (Li-Po) battery and by a mini photovoltaic (PV) panel able to recharge the battery when sufficiently illuminated.

The wireless communication was based on Bluetooth Low Energy 5.0 (BLE) stack, which allows to ensure a good communication range (up to 200 m in line of sight) with very low power consumption (10–500 mW) [32,33] (Figure 1). 

### 2.1. Hardware

From the hardware side, the main innovation with respect to the sensor architecture described in [31] is represented by the implementation of a Micro Electro Mechanical Systems (MEMS) pressure transducer, namely the Honeywell SSCDANN150PAAA3. Among the different models present on the market, this device was chosen for various reasons. The selection of this model was carried out by taking into account the requirements of the monitoring activity and the features of the developed electronic board.

Firstly, the pressure range typical of this application is 0–8 Bar less than the full-scale range of this sensor (150 psi approximately 10,34 Bar). Another feature of this model is that it is an absolute pressure transducer, translating into a more robust solution for this context useful in obtaining reliable diagnostic data. 

The single-axial barbed port of the sensor has the dimensions suitable to host a 4-mm-diameter plastic tube, which allows a safe connection to the wagon test points. Moreover, the I2C analog interface and the 3.3 V supply voltage are essential features for the transducer to be hosted on the developed sensor node. A crucial aspect of the implemented transducer is represented by the current consumption, as already pointed out previously. The chosen analog version of the pressure transducer has a current consumption of 2.1 mA while the serial interface version absorbs approximately 3.1 mA. In the end, another interesting feature is constituted by the power supply. In this case, the transducer is power supplied directly by a digital pin of the microcontroller. Therefore, in this way, the sensor is supplied only when a measure is needed. The measurement procedure takes some milliseconds as is shown in the following, while during the remaining time the current consumption of the pressure transducer is zero. 

The main features of the chosen transducer are summed up in Table 1.

The board’s main power source comes from the solar energy harvested by means of a mini-PV panel, whose main features are reported in Table 2. This energy harvester was chosen to optimize the ratio between compact dimensions (suitable to be host on the top of the sensor board) and output power necessary to recharge the Li-Po battery. The PV panel is connected to the LTC3331 component that performs on board power management. This component implements a Buck-Boost DC/DC converter with an Energy Harvesting Battery Charger, which manages the battery charge using solar energy providing a stabilized voltage supply for the microprocessor and for all on board sensors. The LTC3331 chip does not allow a Maximum Power Point Tracking (MPPT), hence it has been used with the internal hysteretic controller with a proper selection of the voltage thresholds. In order to continuously monitor the sensor node battery status, a LTC2942 battery gauge was inserted. This IC (Integrated Circuit) perform the estimation of the battery State of Charge (SoC) and measures the battery voltage and temperature.

The realized electronic board prototypes are enclosed in 3D printed boxes realized in CPE (Chlorinate Polyethylene), whose characteristics are suitable for the harsh railway environment. The final dimensions of the sensor node box are 91 × 70 × 61 mm, with a weight of 445 g (inclusive of insulating gel filling the box). The hardware functional scheme of the sensor is shown in Figure 1 and a photo of the realized sensor node prototype is shown in Figure 2. 

### 2.2. Software

The sensor node firmware, implemented on the DSPIC33EP512 microcontroller and on the BT840F module, was written with a special attention to the power consumption minimization, which is a crucial aspect of the design. Despite the presence of the mini photovoltaic panel, it is important to save energy due to the difficulty in obtaining a good sun exposure when the sensor is mounted on wagon, or when the PV panels are shielded for any reason during travel (e.g., in long tunnels or underground sections of the railway) in order to acquire data continuously. The software architecture and functionalities are represented in Figure 3.

According to Figure 3, the acquisition software is implemented on two microcontrollers: the BT840F (which controls the communication) and the DSPIC33 (which controls the acquisition). The acquisition procedure starts when the sensor node detects the presence of the data collector gateway on the same BLE sub-network. In order to simplify the connection between the sensor nodes and the data collector, the sensor node was not paired with the data collector but just starts to communicate when the gateway subnetwork is discovered. When this occurs, the BT840F turns off the antenna and wakes up the DSPIC33. The DSPIC33 starts to acquire pressure data continuously and a pressure measure is obtained every second (1 Hz) as a mean over 4 pressure samples at 200 Hz. This mean measures are stored in a buffer for a duration of 20 s. These acquisition parameters (1 Hz sampling frequency and 20 s of acquisition) were selected according to the transient pressure dynamics in the brake system [10,11]. The acquisition phase of a pressure measure lasts approximately 28 ms each second (4 × 5 ms = 20 ms for the acquisition phase +8 ms for other operations) and the microcontroller goes to sleep after this period and it automatically wakes up every second in order to collect the next sample. Data transmission of the entire buffer will occur before the end of 20 s and it takes 3 s approximately. In this way, the antenna is always turned off and it is switched on only when data have to be sent to the gateway. This procedure allows a huge saving in energy consumption since the transmission phase is the most power consuming. An estimation of sensor node power consumption could be obtained using the absorbed currents in Table 3 and using (1).

The proposed sensor nodes were equipped with a 2000 mAh Li-Po battery, so according to the estimated absorbed current, every node is able to work continuously more than two months (67 days) without any inflow current coming from the PV panel considering an active time of 8 h a day.
(1)$$I_{mean}=\frac{I_{p}\cdot t_{p}+I_{\mu }\cdot t_{\mu }+I_{s}\cdot t_{s}+I_{a}\cdot t_{a}}{T_{cycle}}.$$

Data transmitted by sensor nodes to the gateway every 20 s are constituted basically by the pressure measurements and the time when the acquisition has been performed, which is fundamental to correlate data during the postprocessing phase. In addition, sensor nodes transmit useful diagnostic data concerning the battery voltage, the environmental temperature, and an index of the quality communication between them and the data collector (RSSI value).

## 3. Sensor Calibration

A preliminary calibration of the pressure sensors was carried out in laboratory, taking advantage of the available instrumentation, in order to evaluate the sensor nodes performance and to characterize them. The instrument used to produce a pressure reference was a WIKA CPC6050 and the pressure range adopted as reference was −90–90 mBar with 10 mBar steps. The sensor node was connected to the output port of the reference instrument through a pneumatic tube, while measurement data were read and stored by means of serial communication with a laptop (Figure 4).

The adopted calibration procedure involves the following steps. Firstly, a pressure difference Δ*p* was imposed using the reference instrumentation. The reference instrumentation pressure *P_WIKA_* (which includes a very accurate measurement of the atmospheric/barometric pressure) and the sensor node pressure *P_sens_* were collected. This operation was repeated 19 times using a 10 mBar step in the −90–90 mBar pressure range. In the end, the offset *P_off_* characterizing the sensor node measurement was computed according to (2):(2)$$p_{off}=\frac{\sum_{k=1}^{19}P_{WIKA}{\left(\Delta p\right)}_{k}-P_{sens}{\left(\Delta p\right)}_{k}}{19}.$$

The corrected sensor node pressure measurement *p_s_* was then obtained through (3)
(3)$$p_{s}(\Delta p)=p_{sens}(\Delta p)+p_{off}.$$

In Figure 5, an example of calibration data is shown.

## 4. Field Campaign and Experimental Set-Up Description

The developed sensor nodes were tested during a 5-month field campaign (from 11 November 2020 to 20 April 2021) where it was possible to install them on a T3000e freight wagon, made available by Mercitalia Intermodal SpA (Figure 6). This kind of wagon is employed in very long travels for intermodal transportation of both trailers and containers. This situation represents clearly the perfect workbench for the braking system monitoring. The track sections and number of travels run by the instrumented wagon during the field test are summed up in Table 4.

As in the case of other railway vehicles, each T3000e wagon is endowed of some test points placed in suitable position of the braking system that are used during maintenance interventions to perform tests on the braking system efficiency. It seemed the best choice to take advantage of these test points to mount the sensor nodes on the wagon.

The T3000e is composed of two semi-wagons. Three sensor nodes were mounted on the first semi-wagon on the three points of interest of the braking system, namely one on the Main Pipe (MBP), one on the braking cylinder (BC), and the last on the weighing valve (WV). The monitoring of the braking system in these points was chosen accordingly to the diagnostic need, allowing to verify if a wheel is braked when it should not have been. In addition, other diagnostic information concerning wheel flattening, wear of pads, and axle box overheating could be collected. During this experimental campaign commercial, wired sensors were installed on the same test points in the adjacent semi-wagon in order to use the collected data as benchmark. The wired pressure sensor characteristics are listed in Table 5. A scheme of the instrumented wagon is shown in Figure 7.

### 4.1. Test Points and Sensor Node Connections

As already pointed out, it was chosen to take advantage of test points already present on-board the wagon to carry out pressure measurements. Usually during maintenance operations, the instrumentation is connected through tests points by means of the universal connector described in Appendices A.3 of UIC 543-1. Therefore, the idea was to replicate those connectors in order to link safely sensor nodes to test point during the field tests. An example of the test point for the weighing valve and the realized connection is shown in Figure 8.

The three sensor nodes were positioned on the external side of the wagon to guarantee a good sun exposure of the photovoltaic panels. Due to this external position, the sensor nodes were connected with pressure test points through the use of the previous described connectors and of plastic tubes appropriately shielded to avoid any failure related to debris impact. Sensor nodes connected to the braking cylinder (BC) and the weighing valve (WV) are visible in Figure 9a, while the one used to perform measurements on the main brake pipe (MBP) is shown in Figure 9b.

### 4.2. Data Collector and Gateway

Sensor nodes transmit their data wirelessly to a gateway data collector placed on the wagon (white box of Figure 10). The data collector also implements a 4G/LTE gateway to transfer the data on the cloud. This system consists of an industrial pc, a BLE master board to communicate with sensor nodes, a 4G/LTE modem, a GPS receiver, and a custom electronic board developed to manage the power supply of the entire system.

The gateway data collector’s function is to receive data from the wireless nodes through a dedicated BLE board, store them locally, and send them remotely. The gateway is fed through a DC/DC converter and a Lead-Acid battery. This battery is recharged by an axle-box synchronous generator when the train overpasses 10 km/h. The developed custom electronic board was used to implement an energy saving logic based on the speed values acquired from the axle-box generator through CAN communication. Going into detail, the electronic board turns on the industrial PC and the devices connected (BLE board, 4G/LTE modem,…) when a wagon movement is detected while it turns them off when the train does not move for enough time (approximately 5 min). In addition, to save energy and increase the system efficiency, this technique allows the gathering of only useful data when the train is in motion, limiting the amount of data stored on the PC and which is transmitted remotely. The power manager board, in addition, acquires and transfers to the PC via serial communication, and diagnostic data regarding the main status and the speed values come from the axle box.

The acquisition software is started automatically when the PC is active, and after the collection of the pressure measurements, it associates them with the GPS information in order to achieve a complete fleet position monitoring. A scheme of the described system is represented in Figure 11.

## 5. Experimental Results and Data Analysis

During the field campaign, the instrumented wagon traveled mainly on the Verona-Rotterdam track section. This railway track nowadays represents a very important route crossing Europe for goods transportation (Figure 12). 

The field campaign allowed us to test the functioning of the monitoring system in a real case scenario and to collect a good amount of experimental data. Data acquired by the industrial wired sensors were used to validate the measurements performed by the developed wireless sensor nodes. In addition, during the post processing phase, the speed measurement and GPS signals acquired were taken advantage to superimpose the data carried out on different travels, verifying the measurements’ repeatability. Thanks to this possibility, data comparison and superposition results are much more accurate. The availability of this information, moreover, made it possible to correlate the occurring of a braking event with the position of the wagon on the railway track. In the end, an analysis on the possibility of using the collected data to perform a predictive maintenance of the braking plant was carried out. Energy harvester performances during the field campaign were deeply analyzed in paragraph 5.2.

### 5.1. Validation of the Pressure Measurements Acquired by Sensor Nodes

In order to acquire a validation of the developed measurement devices, the first step was the comparison between data acquired by industrial sensors and the ones coming from the wireless sensor nodes. A good coherence between the two datasets can be observed looking at Figure 13, referring to the track section Verona-Rotterdam. In addition, it is possible to appreciate the braking events correlation in terms of pressure in the braking cylinder and train speed trends. As soon as the pressure in the braking cylinder increased, the train speed began to decrease as one would expect. The difference in values between measurements related to the weighing valve of the two systems can be explained by the fact that the two semi-wagons monitored were loaded differently, which often happens. 

To complete the validation procedure, different travels on the same track section (Verona-Rotterdam) were superimposed to verify the measurements repeatability (Figure 14). Data acquired during different travels were realigned considering the speed measurement and GPS information and making the superimposition procedure straightforward. As can be seen, although braking phases can occurr in different situations from time to time, there was a good coherence between braking events performed in different travels, and also speed trends were quite repeatable from travel to travel. This analysis could also be useful to assess the different driving styles used by machinists in various travels on the same track section.

### 5.2. Analysis of the Energy Harvester Performances

An aspect of paramount importance to be carefully analyzed during field tests on wireless sensor nodes is the efficiency of the implemented energy harvesting system. This aspect represents a key feature of the developed device since it allows us to make sensor nodes almost independent from an energetic point of view by any wired power supply. Laboratory tests are surely useful to check, in a preliminary stage, that the developed board works properly on the electrical side. However, it is nearly impossible to reproduce, in a confined environment, all the possible threats that can undermine the efficiency of the harvester (cloudy weather, tunnels, dust, debris impact, presence of few solar hours, temperature variation). Only field tests can therefore assure that the chosen harvester is suitable for the monitoring purposes of the specific application (in terms of scenario and duty cycle). Some data collected during the field tests are analyzed in the following.

In most cases, the balance between the inflow energy coming from the PV panel and the very low consumption offered by the implemented acquisition logic and by the use of ultra-low-power electronic components allows sensor nodes to perform continuous measurements without discharging batteries. In Figure 15, the battery voltage time history of the three sensor nodes during a travel on the Verona-Rotterdam track section is represented as example of the performance. The data were acquired after 4 months of experimentation, and so it is also possible to see the aging effect on batteries.

In particular, the voltage trend of the three sensors kept almost constant during several hours of working activities, witnessing the efficiency of the power consumption logic implemented and of the energy harvester employed. The blank voids are due to the fact that, as already pointed out, when the train stops for a certain amount of time (that happens in case of certain crossings or stations), the whole system is put to sleep to save energy. However, some limitations of the energy harvester employed came out for this application. An example of this situation is visible in Figure 15.

The battery voltage curve of sensor nodes E5 and E0 (blue and red curve respectively) showed a steep decreasing trend during night hours, reaching values under the 3V threshold. This condition represents almost the limit for sensor nodes to stay awake and active for acquisition purposes. The reason for this situation can be found in some limitations of the implemented energy harvesters for the specific application. In particular, the main cause could be that, from the experience collected in the field test, freight trains travel very often at night, which translates in the system operating clearly with no sunny conditions. If the train is made to stop in railway depots, the PV panels could be hidden by adjacent convoys or by the structure of the depot itself. Therefore, when the instrumented convoy begins a new journey, the sensor nodes might not be sufficiently charged to run acquisitions, at least initially especially after some months of installation. Other factors that, resulting from the experimental activity, can undermine the PV panels’ efficiency are the presence of very long tunnels on the traveled track sections and the presence of dust covering the external layer of panels. Anyway, as can be appreciated from Figure 16, as soon as the sunlight hits the PV panels, the developed electronics were able to recharge the batteries in an efficient way, allowing the sensor nodes to perform their acquisition task with aged batteries.

### 5.3. Diagnostic Analysis

The collected data could be used for predictive maintenance of the braking systems. In particular, the pressure measures could be used to describe the state of health of the brake system. A complete pressure time-history related to a travel carried out on the Verona-Rotterdam is shown in Figure 17a. As can be observed from the pressure curves of main pipe and braking cylinder, several braking events occurred during the travel. The focus was then moved on to single braking (Figure 17b). It can be easily noted that the sampling frequency adopted for acquisition is enough to describe the braking pressure dynamics with good accuracy the.

Analyzing Figure 17a, it is possible to carry out some considerations regarding possible applications of a predictive maintenance approach [1]. Some approaches mainly based on machine learning [34,35] are already available in the literature, however the innovative solution proposed in this paper was based on the smart measurement of pressure values in some peculiar points of the systems to identify possible faults in the main components. This way it could be possible to run simple and efficient algorithm directly on the on-board gateway in real time to perform these diagnostic activities. Some examples of this approach are proposed in the following.

Referring to Figure 17b, when the braking phase begins, a decreasing in pressure values at the main pipe was observed and there was a contemporary pressure increase in the braking cylinder. If this does not happen, it would mean that a fault on the monitored braking cylinder may have happened, and that bogie is not able to brake. Moreover, faulty conditions of the brake cylinder can be identified through the analysis of the duration of the braking phase, since its lasting in time is due to the type of braking mode selected before the travel (usually the selection is taken between freight mode and passenger mode though a specific mechanical leverage). If the time duration of the braking phase in not coherent with the selected mode (and prescribed by the UIC540 standard), some problems occurred.

Regarding the magnitudes of pressure, the one at braking cylinder is a function of the wagon load and it is managed by the weighing valve. Clearly, the higher the loading, the higher the brake force requested. This higher braking request requires higher pressure values at the braking cylinder. Therefore, monitoring the weighing valve pressure and the brake cylinder pressure, it is possible to understand if there is a malfunctioning of the weighing valve component. In addition, the monitoring of the weighing valve pressure can be used to distinguish the loaded or unloaded wagon condition. To appreciate the effectiveness of this method, two journeys with similar train speed profiles were taken into account. Looking at Figure 18, it is possible to underline that the braking phases of the unloaded wagon on travel 18th of March were less severe than the ones performed by the loaded wagon during travel 4th of February. In conclusion, it could be possible to run a diagnostic algorithm based on the previous considerations in real time on the on-board PC in order to perform continuous monitoring of the system useful for a predictive maintenance approach.

## 6. Conclusions

The development of an innovative monitoring system composed of wireless sensor nodes aimed at performing pressure measurements on freight train braking systems was proposed. The choice of the electronic components and the implementation of a smart firmware were carried out to minimize the power consumption. Thanks to this software implementation and to the adoption of a suitable energy harvester, the developed sensor nodes result as autonomous from an energetic point of view.

The designed system was installed on a freight wagon for a 5-month field test. The set-up was arranged to perform measurements in specific points of the plant, realizing safe mechanical connection to the existing vehicle brake system test points. During the field test, the wagon traveled more than 24,000 km on important track sections across Europe and the correct functioning of the system was verified. 

In the post-processing phase, an initial focus was devoted to check the efficiency of the implemented energy harvester in recharging sensor nodes batteries and to verify the consistency of pressure measurements through the comparison with wired commercial sensors. The use of the acquired data for diagnostic purposes of the brake system was then validated through a deep analysis of the experimental data. It was demonstrated how a predictive maintenance approach can be employed taking advantage of the monitoring of the pressure dynamics in some peculiar points of the plant. Through the development of specific algorithms to be run on-board, the system makes it possible to identify the presence of faults in the braking system, reducing the cost and time related to unexpected breakdowns and increasing the reliability of this transportation system.

## Figures and Tables

**Figure 1 sensors-22-01876-f001:**
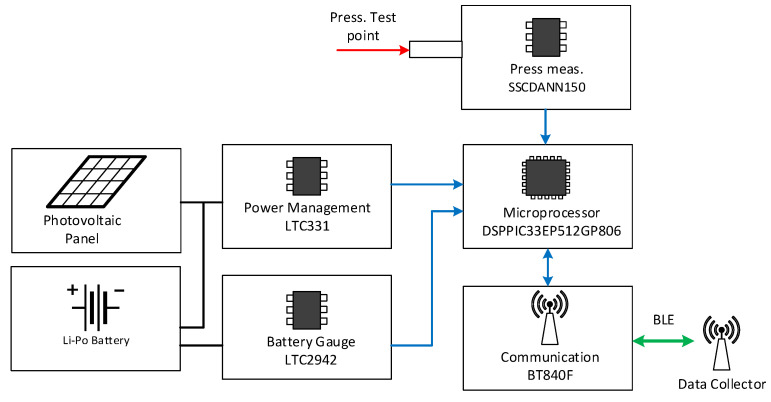
Hardware architecture.

**Figure 2 sensors-22-01876-f002:**
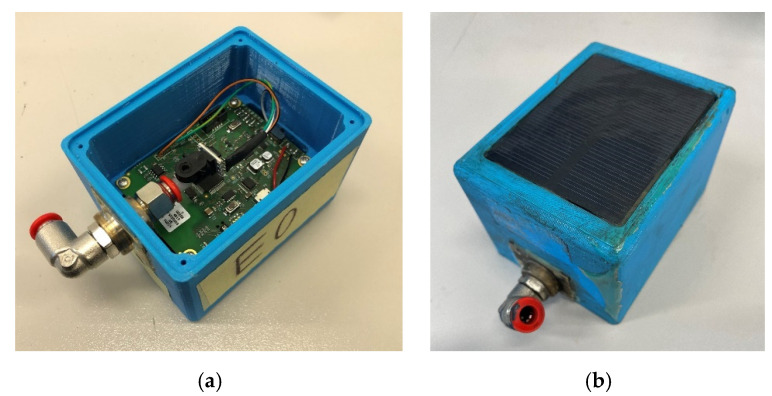
(**a**) Internal view of the sensor node; (**b**) external view of the sensor node showing the PV panel installed on the top of the box.

**Figure 3 sensors-22-01876-f003:**
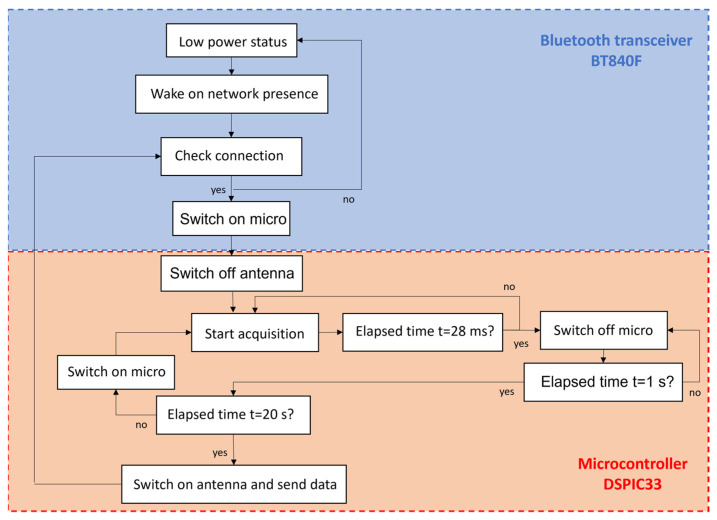
Software architecture and functionalities. The software is implemented on two microcontrollers: BT840F (blue part) and DSPIC33 (red part).

**Figure 4 sensors-22-01876-f004:**
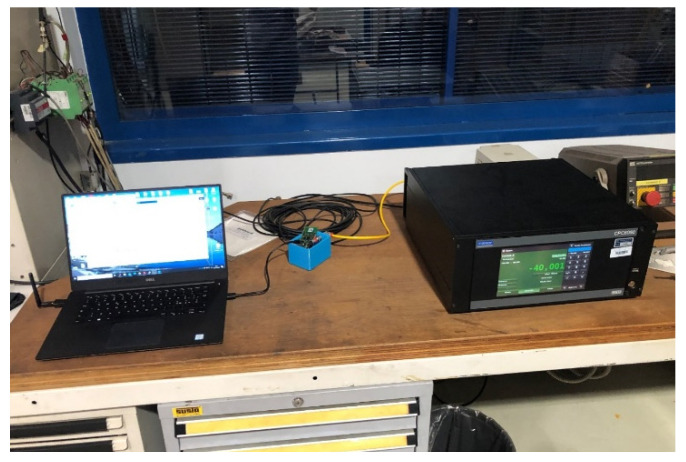
Calibration test bench.

**Figure 5 sensors-22-01876-f005:**
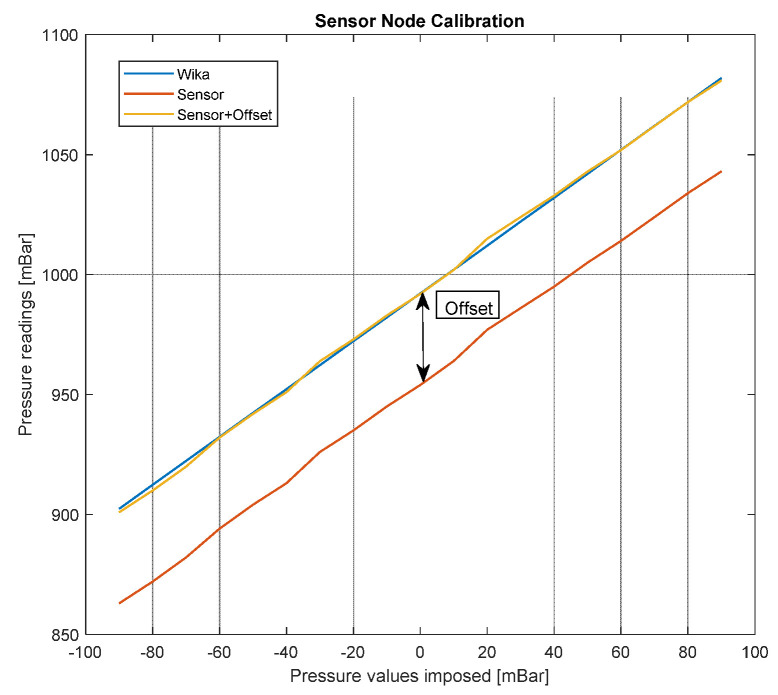
Example of calibration experimental data.

**Figure 6 sensors-22-01876-f006:**
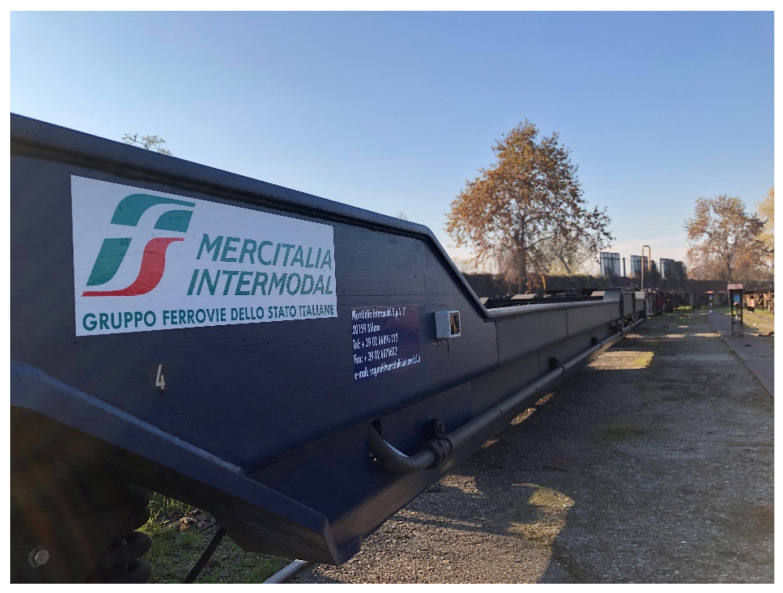
Mercitalia Intermodal T3000e freight wagon.

**Figure 7 sensors-22-01876-f007:**
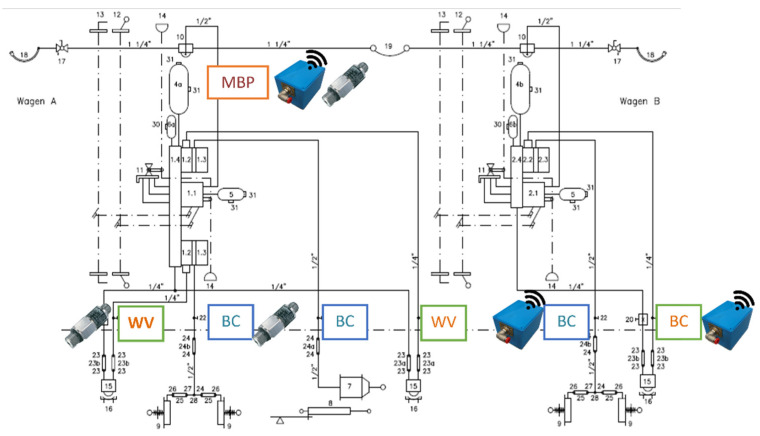
Position of sensor nodes on the Mercitalia Intermodal T3000e wagon.

**Figure 8 sensors-22-01876-f008:**
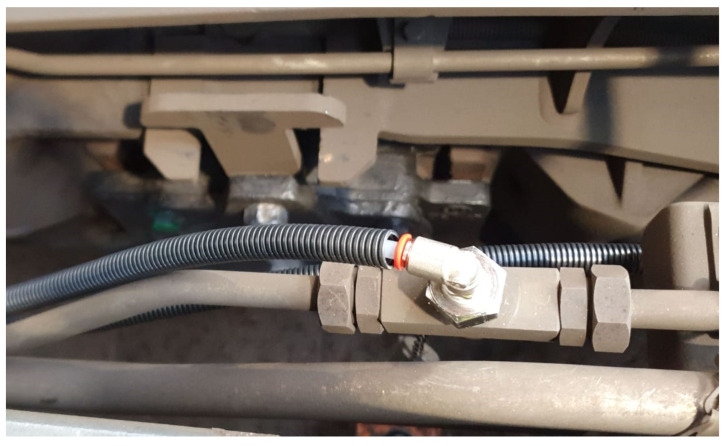
Mechanical connections.

**Figure 9 sensors-22-01876-f009:**
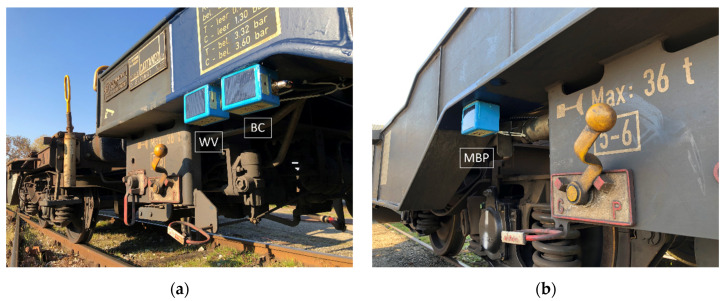
Position of sensor nodes on Mercitalia Intermodal wagon (Figure 7). (**a**) Node BC and WV (**b**) Node MBP.

**Figure 10 sensors-22-01876-f010:**
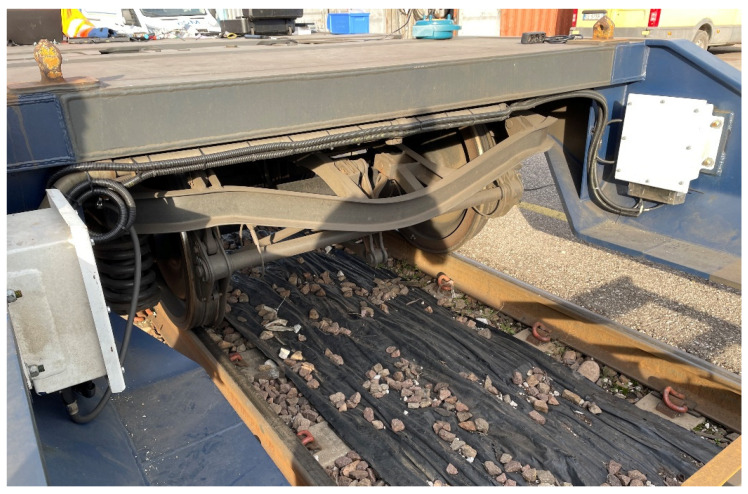
Data collector box on Mercitalia Intermodal wagon.

**Figure 11 sensors-22-01876-f011:**
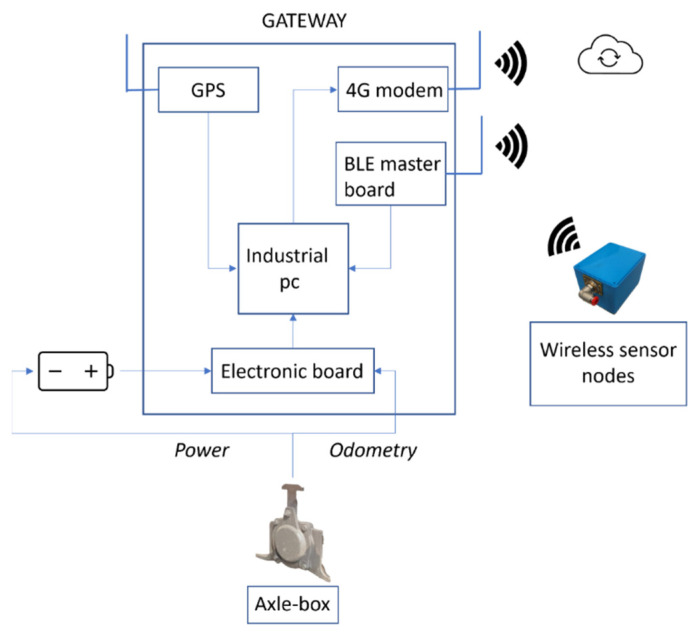
Data collector architecture.

**Figure 12 sensors-22-01876-f012:**
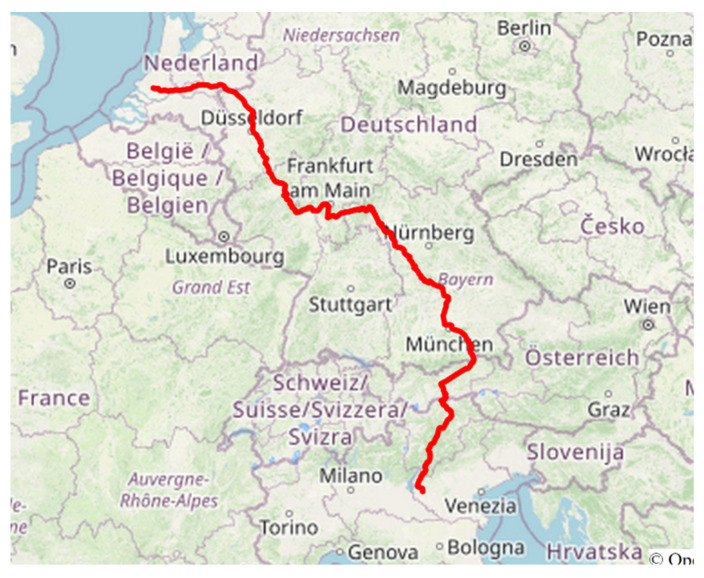
Example of GPS track data during field experiments.

**Figure 13 sensors-22-01876-f013:**
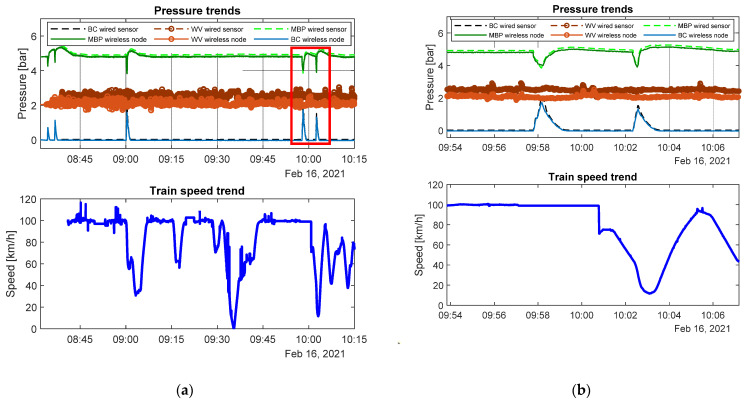
Example of comparison of pressure data. (**a**) Pressure data during travel; (**b**) example of a braking event.

**Figure 14 sensors-22-01876-f014:**
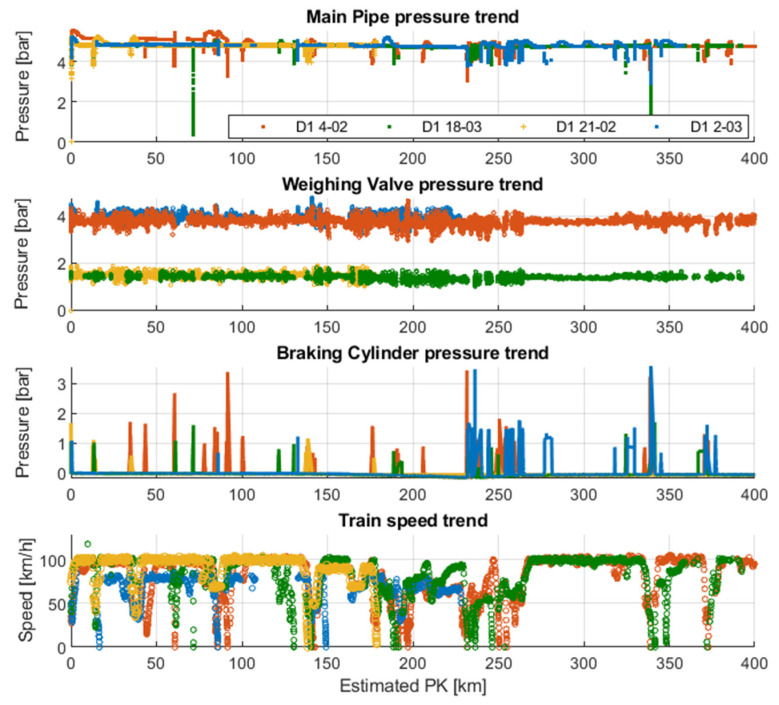
Example of collected pressure data during four different travels (4 February, 21 February, 2 March, 18 March) on the Verona-Rotterdam track.

**Figure 15 sensors-22-01876-f015:**
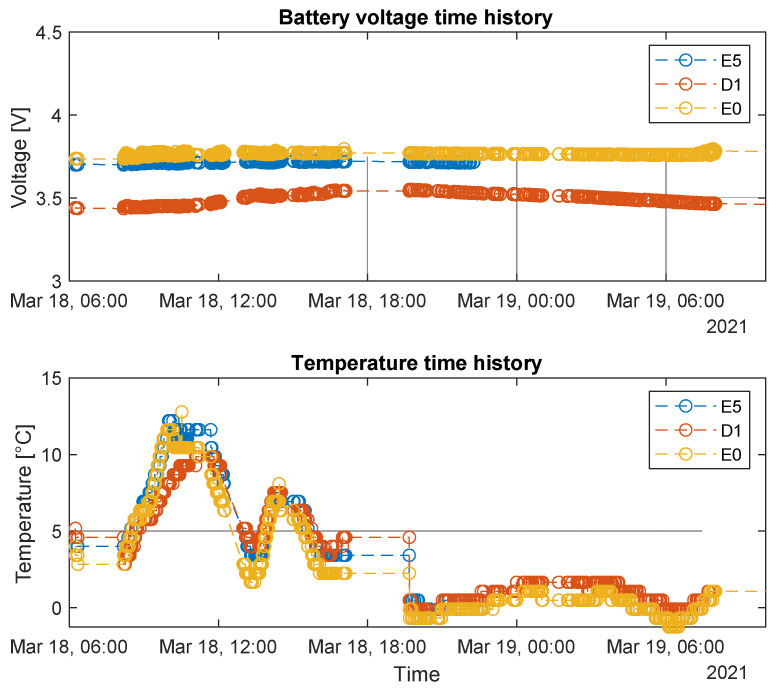
Example of battery voltage and temperature data.

**Figure 16 sensors-22-01876-f016:**
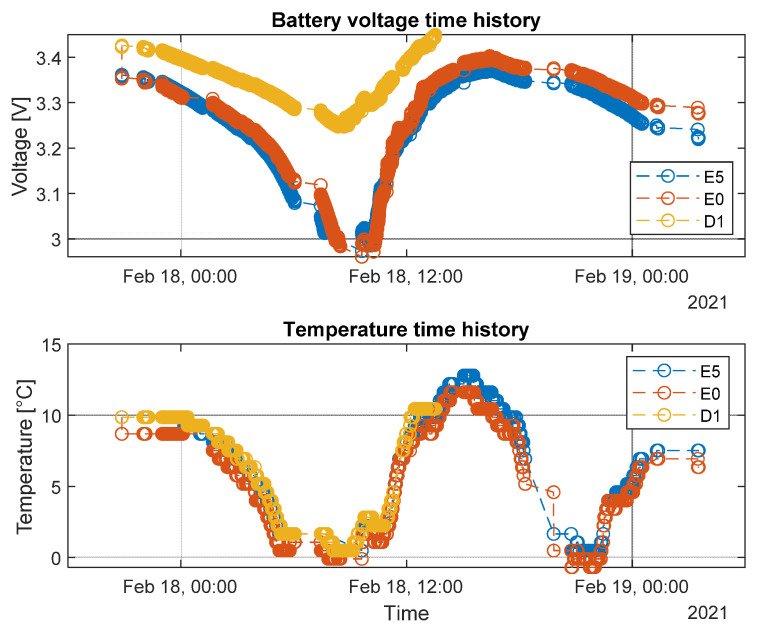
Example of battery voltage and temperature data.

**Figure 17 sensors-22-01876-f017:**
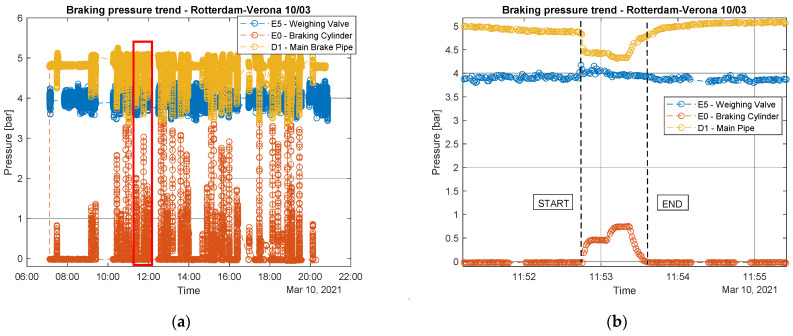
Pressure measures during a wagon travel (**a**) Pressure history (**b**) Pressure variation during a braking event.

**Figure 18 sensors-22-01876-f018:**
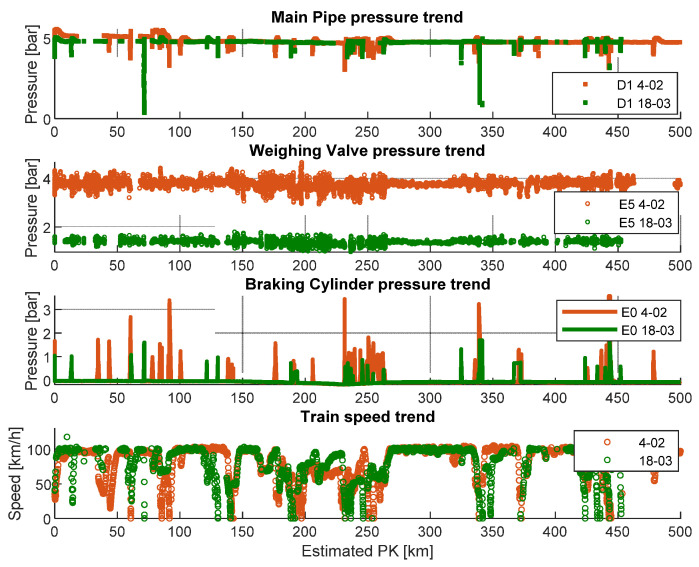
Comparison of braking events measures on different brake system points during two-wagon travel. Red lines refer to 4 February’s journey (loaded wagon), and green lines refer to 18 March’s journey (non-loaded wagon).

**Table 1 sensors-22-01876-t001:** Pressure sensor characteristics.

SSCDANN150PAAA3 Characteristics	
Sensor Type	Absolute
Pressure Range	0–150 Psi
Interface	Analog (0.33 V~2.97 V)
Resolution	12 bit (internal ADC)
Operating Temperature	0–50 °C
Supply Voltage	3.3 V
Current consumption	2.1 mA

**Table 2 sensors-22-01876-t002:** Photovoltaic panel characteristics.

PV SEEED 313070004 Characteristics	
Material	Monocrystalline Sylicon
Typical peak power	0.55 W
Open circuit voltage power	8.2 V
Voltage at peak power	5.5 V
Current at peak power	100 mA
Length	70 mm
Width	55 mm
Weight	17 g
Efficiency	17%

**Table 3 sensors-22-01876-t003:** Current consumption.

	Pressure Sensor (I_p_)	Micro (I_μ_)	Sleep Phase (I_s_)	Commun. (I_a_)	Total (I_cyle_)
Mean curr. (mA)	1.7	7.24	0.69	8	3.72
Time (s)	20	0.020 every 1 s	0.98 every 1 s	3	20

**Table 4 sensors-22-01876-t004:** Travels during field test.

Track Section	Estimated Distance (km)	Number of Travels
Milano-München	610	2
München-Milano	610	2
Verona-München	450	2
München-Verona	450	2
Verona-Rotterdam	1120	9
Rotterdam-Verona	1120	9
Total	24,400	26

**Table 5 sensors-22-01876-t005:** Pressure sensor characteristics.

Wired Pressure Sensors Characteristics	
Sensor Type	Relative
Pressure Range	0–10 Bar
Precision	0.1%
Interface	CANOpen
Operating Temperature	−40–125 °C
Supply Voltage	8–32 V

## Data Availability

Not applicable.

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
