# Peer review of "Energy Autonomous Wireless Sensor Nodes for Freight Train Braking Systems Monitoring"

_sensors, 2022, doi:10.3390/s22051876_

Round 1

Reviewer 1 Report

This work presented a practical realization of energy-autonomous wireless sensor for train braking system monitoring with on-field tests. The work is interesting with practical application realization. A couple of the problems/suggestions can be found below. 

At the end of the literature review, it was summarized that the related technique is feasible to realize practical condition monitoring. But what problem that has not been solved in this engineering field was not clarified. The 'gap' should be well summarized before introducing the work presented in this paper. What is the meaning to fill this 'gap'? This will help readers to follow exactly what is going to be solved as a new contribution. 

The work presented detailed application precess regarding how to operate the sensor. This is a good engineering report but not a clear presentation of research work. To me, a research paper should concentrate more on the novel idea/technique/design/progress that is developed in the current work. Therefore, the realization energy-autonomous, the selection, set up, and arrangement of photovoltaic panels, as well as any new design of the sensor should be highlighted with much more details, while the descriptions of the general hardware and operation process, like the sensor calibration and data acquisition, can be shortened. In addition, in the results section, 

Reviewer 2 Report

Dear Authors, 
congratulations for the work done.
The paper aims at presenting an energy autonomous wireless sensor nodes (WSN)-based system for monitoring the health status of freight train braking systems (based on pressure data of a freight wagon braking plant during a 5-month long field test), which are one of the more critical element in case of failure. 
Results show that the proposed self-powered system is usefull because: (1) in this type of applicaiton it is very difficult to apply wiring, (2) the power supply system is efficient and durable, (3) the algorithms of the monitoring system allow energy-saving and efficient data transmission, (4) the monitoring system allows recognizing the level of load of a train, and (5) the collected data can be used to carry out continuous fault detection, and could be used for predictive maintenance.

In general, the paper is interesting and well written. The following comments have been provided to improve the readablity of the paper:
1. Abstract: It should be better to add more detail/quantify the results of the esperiments in this part of the paper.
2. Line 54: The term "..." can be substituted with "etc.".
3. Line 90: You may also consider the following studies to extend the literature review:
- https://doi.org/10.3390/a13100254
- https://doi.org/10.1016/j.nanoen.2017.05.018
- https://doi.org/10.1016/j.enconman.2021.114413
- https://doi.org/10.1016/j.measurement.2021.109856
- https://doi.org/10.1016/j.ymssp.2021.108113

4. Line 114: Please, replace "Section 2 describes the design the wireless.." with "Section 2 describes the design of the wireless..".
5. Line 128: It would be better to quantify the parameters you mentioned. For example, ranges for "good" communication range and "low" power consumption can be provided.
6. Figure3: A legend is needed, please.
7. Expressions 2 and 3: Please, explain/define all the symbols here used. Are "Psensor" and "Psens" the same parameter?
8. How many sensor nodes did you install during the field test? It seems that the wireless sensor network (WSN) consists of three units. Is this correct? Does each WSN powered using one PV panel?  Is this correct? What are the characteristics of the PV panel/power system that fed each WSN? and where did you install the PV panel? 
9. Line 387: Please, replace "senor" with "sensor".
10. Figure 17: Please, use different colors for the different curves. 
11. It would be interesting to provide an example of solution for the predictive maintenance allowed by the proposed monitoring system.

Best regards.

Round 2

Reviewer 1 Report

The authors addressed my previous comments, and the work could be considered for publication now.